# Pigmentation level of human iPSC-derived RPE does not indicate a specific gene expression profile

Yoko Nakai-Futatsugi[1,2,3]*, Jianshi Jin[4†], Taisaku Ogawa[4], Noriko Sakai[1,2], Akiko Maeda[1,3,5], Ken-ichi Hironaka[6], Masakazu Fukuda[6], Hiroki Danno[6], Yuji Tanaka[1], Seiji Hori[2], Katsuyuki Shiroguchi[4], Masayo Takahashi[1,3,5,7]

[1]Laboratory for Retinal Regeneration, RIKEN Biosystems Dynamics Research (BDR), Kobe, Japan; [2]VC Cell Therapy Inc, Kobe, Japan; [3]Ritsumeikan University, Shiga, Japan; [4]Laboratory for Prediction of Cell Systems Dynamics, RIKEN Center for Biosystems Dynamics Research (BDR), Suita, Japan; [5]Kobe City Eye Hospital, Department of Ophthalmology, Kobe, Japan; [6]Knowledge Palette, Inc, Kawasaki, Japan; [7]Vision Care, Inc, Kobe, Japan

*For correspondence:
futatsugi@vcct.jp

Present address: [†]State Key
Laboratory of Integrated
Management of Pest Insects and
Rodents, Institute of Zoology,
Chinese Academy of Sciences,
Beijing, China

Competing interest: See page
11

Reviewing Editor: Ivan Velasco,
Universidad Nacional Autónoma
de México, Mexico

**Abstract** Retinal pigment epithelium (RPE) cells show heterogeneous levels of pigmentation when cultured in vitro. To know whether their color in appearance is correlated with the function of the RPE, we analyzed the color intensities of human-induced pluripotent stem cell-derived RPE cells (iPSC-RPE) together with the gene expression profile at the single-cell level. For this purpose, we utilized our recent invention, Automated Live imaging and cell Picking System (ALPS), which enabled photographing each cell before RNA-sequencing analysis to profile the gene expression of each cell. While our iPSC-RPE were categorized into four clusters by gene expression, the color intensity of iPSC-RPE did not project any specific gene expression profiles. We reasoned this by less correlation between the actual color and the gene expressions that directly define the level of pigmentation, from which we hypothesized the color of RPE cells may be a temporal condition not strongly indicating the functional characteristics of the RPE.

## eLife assessment

This **useful** work describes a novel microscopy-based method to correlate the degree of pigmentation with the gene expression profile of human-induced pluripotent stem cell-derived Retinal Pigmented Epithelial (iPSC-RPE) cells at the single cell level. The presented evidence is **solid** in showing that there is heterogeneous gene expression in iPSC-derived RPE cells, and there is no significant correlation with the pigmentation. By analyzing the expression of some genes related to function, lysosomal- and complement-related pathways were partially enriched in darker cells. This methodology can be used by other researchers interested in analyzing gene expression related to microscopic images.

## Introduction

RPE is a layer of cells paved by hexagonal, brown-pigmented cells, which locates between the neural retina and the choroid. RPE cells play an important role in maintaining the visual system by supplying nutrition to the photoreceptors, phagocytosing mature outer segments of the photoreceptors, facilitating visual/retinoid cycle to produce a photosensitive derivative of vitamin A, absorbing stray light for the vision, and secreting vascular endothelial growth factor (VEGF) for the maintenance of choroid

**eLife digest** The backs of our eyes are lined with retinal pigment epithelial cells (or RPE cells for short). These cells provide nutrition to surrounding cells and contain a pigment called melanin that absorbs excess light that might interfere with vision. By doing so, they support the cells that receive light to enable vision. However, with age, RPE cells can become damaged and less able to support other cells. This can lead to a disease called age-related macular degeneration, which can cause blindness.

One potential way to treat this disease is to transplant healthy RPE cells into eyes that have lost them. These healthy cells can be grown in the laboratory from human pluripotent stem cells, which have the capacity to turn into various specialist cells. Stem cell-derived RPE cells growing in a dish contain varying amounts of melanin, resulting in some being darker than others. This raised the question of whether pigment levels affect the function of RPE cells. However, it was difficult to compare single cells containing various amounts of pigment as most previous studies only analyzed large numbers of RPE cells mixed together.

Nakai-Futatsugi et al. overcame this hurdle using a technique called Automated Live imaging and cell Picking System (also known as ALPS). More than 2300 stem cell-derived RPE cells were photographed individually and the color of each cell was recorded. The gene expression of each cell was then measured to investigate whether certain genes being switched on or off affects pigment levels and cell function.

Analysis did not find a consistent pattern of gene expression underlying the pigmentation of RPE cells. Even gene expression related to the production of melanin was only slightly linked to the color of the cells. These findings suggests that the RPE cell color fluctuates and is not primarily determined by which genes are switched on or off. Future experiments are required to determine whether the findings are the same for RPE cells grown naturally in the eyes and whether different pigment levels affect their capacity to protect the rest of the eye.

blood vessels (*Kim et al., 2013*; *Lehmann et al., 2014*; *Lin et al., 2022*). Abnormalities in the RPE cause a wide variety of retinal degenerative diseases such as age-related macular degeneration (AMD) which is a devastating disease leading to blindness. Medication for AMD is limited, and there are increasing attention on cell transplantation of RPE for the treatment. RPE transplantation for AMD to replace degenerated RPE with fetal RPE was introduced almost three decades ago (*Algvere et al., 1994*). Later, transplantation of autologous peripheral RPE to the degenerated site has also been reported (*Binder et al., 2002*). Our laboratory has pioneered in the production of RPE cells from iPSC and ocular transplantation of iPSC-derived RPE cells (iPSC-RPE) to replace degenerated RPE of AMD patients (*Mandai et al., 2017*; *Sugita et al., 2020*). For quality management of iPSC-RPE transplants, we routinely verify the expressions of key factors such as *RPE65*, *Bestrophin*, *CRALBP,* and *MERTK* by quantitative PCR; PAX6 and microphthalmia transcription factor (MiTF) by immunostaining; and pigment epithelium-derived factor glycoprotein (PEDF) and VEGF by enzyme-linked immunosorbent assay (ELISA). The morphology and pigmentation are also qualitatively checked, but there has been no means to link these appearances with gene expression.

Our recent invention, ALPS, enables the characterization of a cell that is actually showing a behavior of interest (*Jin et al., 2023*). While monitoring the cells in live, ALPS enables to pick up one single-cell for further manipulation. In the present study, we utilized this system to monitor the color of iPSC-RPE cells, then to pick up one of the cells, followed by RNA-sequencing analysis to profile the gene expression of that single-cell. In this way, we concluded the color of iPSC-RPE cells did not project any specific gene expression profile. We reasoned this by analyzing the correlation between the color and the expression level of each gene. We found the genes that define pigmentation itself had less correlation with the color, suggesting the color in appearance is a temporal condition that does not directly represent the function of the RPE.

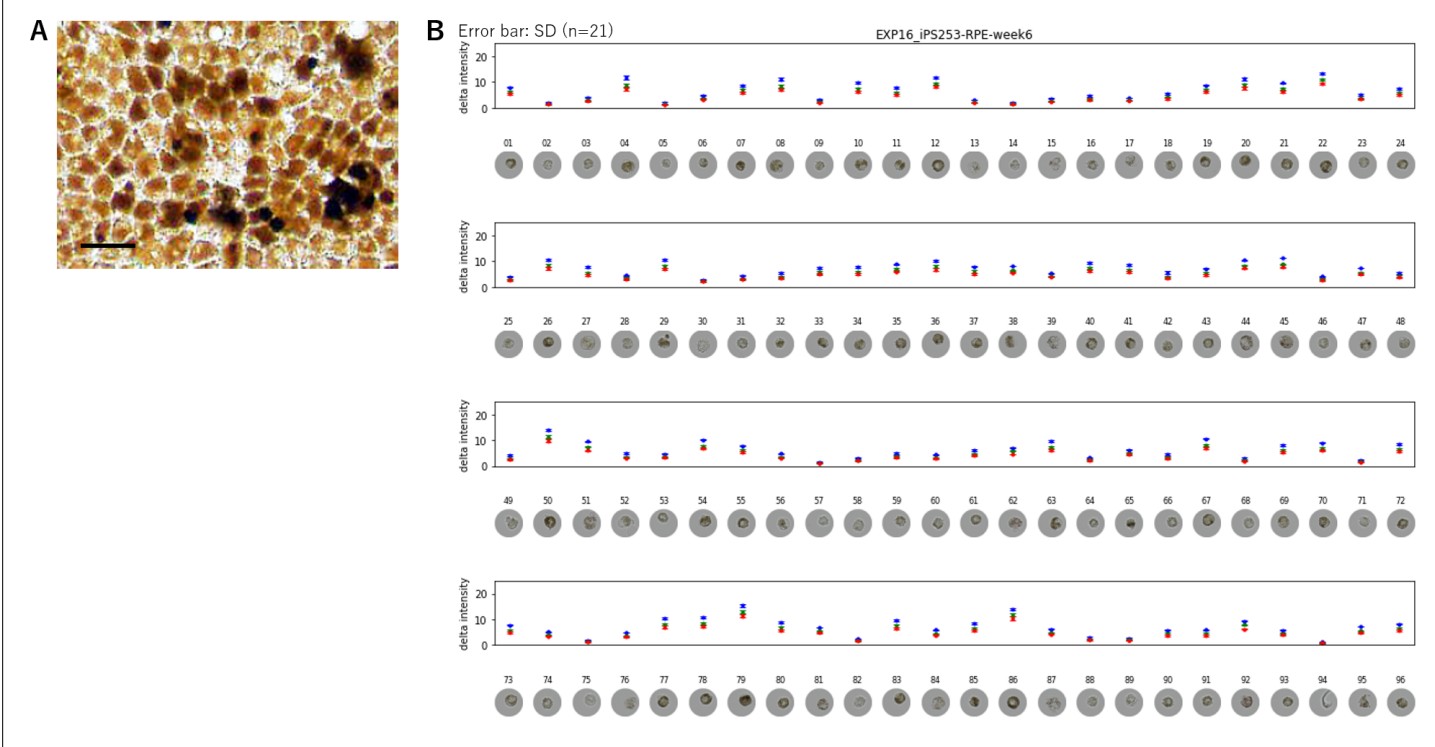

**Figure 1.** Automated Live imaging and cell Picking System (ALPS) enabled photographing each cell before RNA extraction. (**A**) Human-induced pluripotent stem cell-derived RPE cell (iPSC-RPE) in vitro shows a diverse degree of pigmentation among the cells. Scale bar = 20 μm. ( **B**) Each cell was photographed under a light microscope and picked by ALPS. The intensities of red, green, and blue were measured. As there was no difference among the three intensities, the intensity of blue wavelength was used to represent the color intensity of each cell (see also *Figure 1—video 1*).

The online version of this article includes the following video and source data for figure 1:

**Source data 1.** The cells picked by Automated Live imaging and cell Picking System (ALPS) and subjected to single -cell transcriptome analysis.

**Figure 1—video 1.** Since the red-, green-, and blue-intensities were linearly correlated among the human-induced pluripotent stem cell-derived RPE cells (iPSC-RPE), the intensity of blue wavelength was used to represent the color intensity of each cell in the following analyses.

https://elifesciences.org/articles/92510/figures#fig1video1

## Results

### ALPS enabled photographing each cell before RNA extraction

iPSC-RPE cells were produced from two human iPSC-lines of the Center for iPS Cell Research and Application (CiRA; iPSC-lines 201B7 and 253G1). The quality of the generated iPSC-RPE cells were verified by the expressions of *RPE65*, *Bestrophin*, *CRALBP,* and *MERTK* by quantitative PCR; PAX6, MiTF, and Bestrophin by immunostaining; and PEDF and VEGF by ELISA. Hexagonal morphology and pigmentation were also confirmed qualitatively (*Figure 1A*). Normally the degree of pigmentation of RPE cells increases during the culture period, culminating when the cells become confluent, and returns less pigmented when they are replated at sparse density. It is also normal that even at the most confluent state, RPE cells show diverse degrees of pigmentation (*Figure 1A*). To characterize the cells with different degrees of pigmentation, 2304 cells of iPSC-RPE (96 cells each from two different dishes of 12 independent cultures of iPSC201B7-RPE and iPSC253G1-RPE cultured for 6 and 12 weeks; *Figure 1—source data 1*) were photographed under a light microscope (*Figure 1B* photos) and picked as single-cells by ALPS. The intensities of red, green, and blue channels were measured for each cell (*Figure 1B* plots). Because of the strong correlation (*r*>0.99) between the three channels (*Figure 1—video 1*), we used the intensity of the blue channel, or the average intensity of the three channels (so-called brightness), as a proxy for the color intensity of each cell.

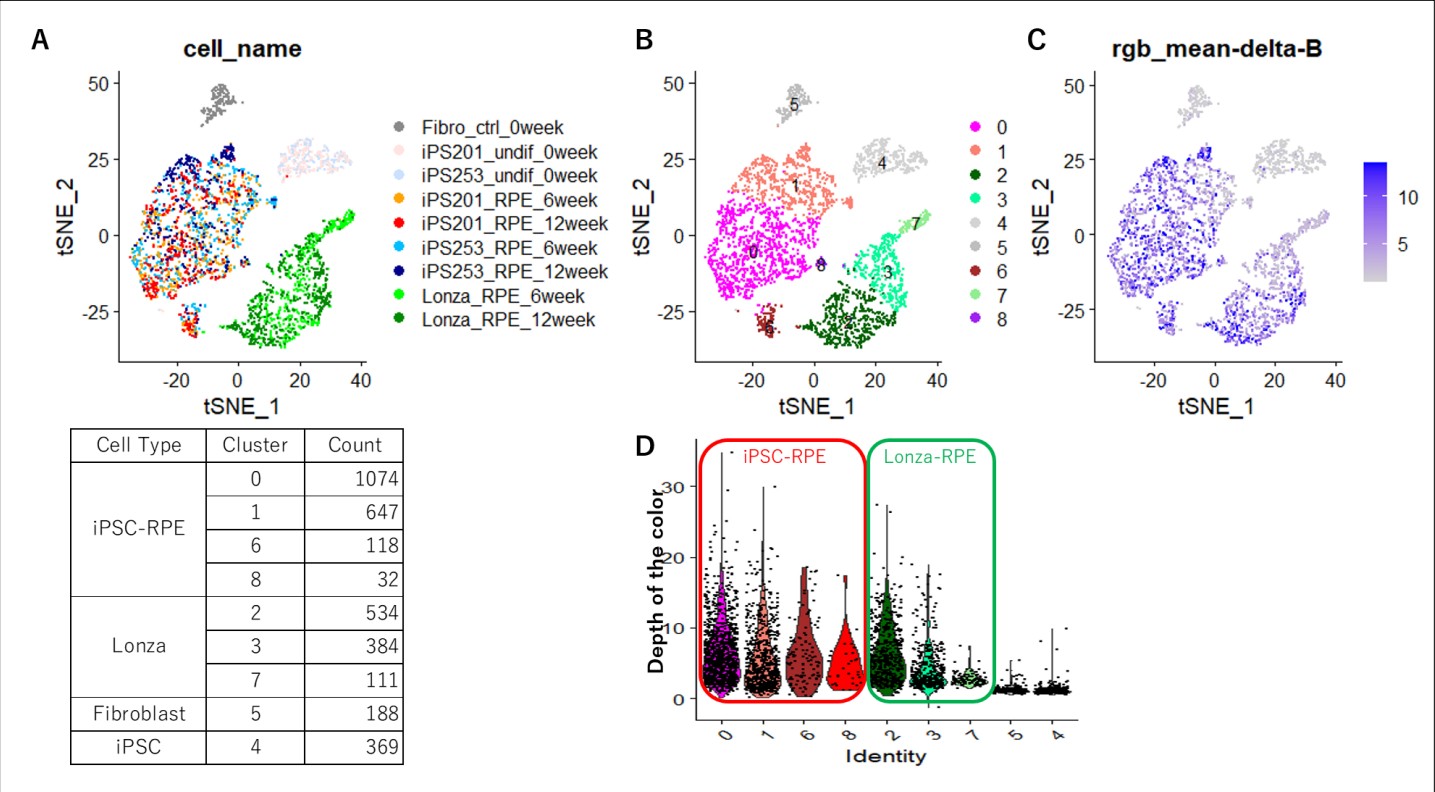

**Figure 2.** Single-cell transcriptome analysis of the cells picked with Automated Live imaging and cell Picking System (ALPS). Two-dimension plotting of the gene expression profile of each cell colored by the cell types (**A**), clusters defined by gene expression profiles (**B**) or color intensities (**C**), and a violin plot quantitatively showing color intensities of the cells in each cluster (**D**). Human-induced pluripotent stem cell-derived RPE cells (iPSC-RPE) formed a different cluster from Lonza-retinal pigment epithelium (RPE). (**B**) iPSC-RPE cells were a heterogeneous population consisting of four sub-clusters. (**C**, **D**) Either very black or very white cells did not localize to a specific sub-cluster of iPSC-RPE.

The online version of this article includes the following figure supplement(s) for figure 2:

**Figure supplement 1.** The expression levels of retinal pigment epithelium (RPE-) and iPSC-markers in human-induced pluripotent stem cell-derived RPE cells (iPSC-RPE) are shown by violin plots of each cluster defined in *Figure 2B*.

**Figure supplement 2.** Gene enrichment analysis of the two major clusters of human-induced pluripotent stem cell-derived RPE cell (iPSC-RPE) (cluster 0 and 1) defined in *Figure 2B* did not elucidate the qualitatively significant difference between the two.

**Figure supplement 3.** A minor cluster of human-induced pluripotent stem cell-derived RPE cell (iPSC-RPE) defined in *Figure 2B*, which consisted of 32 cells (*upper right*), showed a stem cell-like feature with higher expression of proliferation marker *MIK67* (*upper left*) and active cell cycle by gene enrichment analysis (*lower panel*), but was seemingly not a residual of undifferentiated iPSCs as shown by little expression of pluripotency marker *LIN28A* (*upper middle*).

## Dark or white iPSC-RPE cells did not form a specific cluster by single-cell transcriptome analysis

The iPSC-RPE cells photographed and picked by ALPS were then subjected to RNA extraction followed by single-cell RNA sequencing analysis. To compare the gene expression profiles of our iPSC-RPE lines with other commercially available RPE cells, human fetal primary RPE cells purchased from Lonza were photographed and picked by ALPS after cultured for 6 and 12 weeks as well (*Figure 1—source data 1*). The *t*-distributed stochastic neighbor embedding (*t*-SNE) plot of the transcriptomes of iPSC-RPE cells, Lonza-RPE cells together with undifferentiated human iPSCs and human fibroblasts showed our iPSC-RPE and Lonza-RPE in different clusters (*Figure 2A and B*). Although the parameters for *t*-SNE were adjusted to have all the iPSC-RPEs (iPSC201B7-RPE and iPSC253G1-RPE cultured for 6 and 12 weeks; *Figure 1—source data 1*) mingle regardless of the original iPSC-line or the culture period (*Figure 2A*), our iPSC-RPE was divided in four sub-clusters (*Figure 2B*). The major 2 clusters (cluster-0 and –1) did not show critical differences in the expressions of key factors of the RPE (*Figure 2—figure supplement 1*) or the gene-ontology terms (*Figure 2—figure supplement*

2). Interestingly, cluster-8 consists of 32 cells showed the expression of proliferation marker *MKI67* which was not expressed in other clusters besides cluster-4 that represented iPSC (*Figure 2—figure supplement 3*). Unlike iPSCs in cluster-4, the iPSC-RPE cells of cluster-8 did not show the expression of an iPSC-marker *LIN28A* (*Figure 2—figure supplement 3*) denying the possibility of cluster-8 being a residual of iPSCs after RPE differentiation. It could be possible that cluster-8 implies the existence of an adult stem cell population of the RPE.

When the *t*-SNE plot was displayed with the color intensity of each cell, either deeply or lightly pigmented cells did not localize to a particular cluster (*Figure 2C*). Violin plots quantifying color intensities of the cells in each cluster showed all clusters having both dark- and light-colored cells (*Figure 2D*). Interestingly, Lonza-RPE showed biased distribution of cell color among the three clusters (cluster-2,–3, and –7), having highly pigmented cells in cluster-2 and –3 but not in –7 (*Figure 2C*

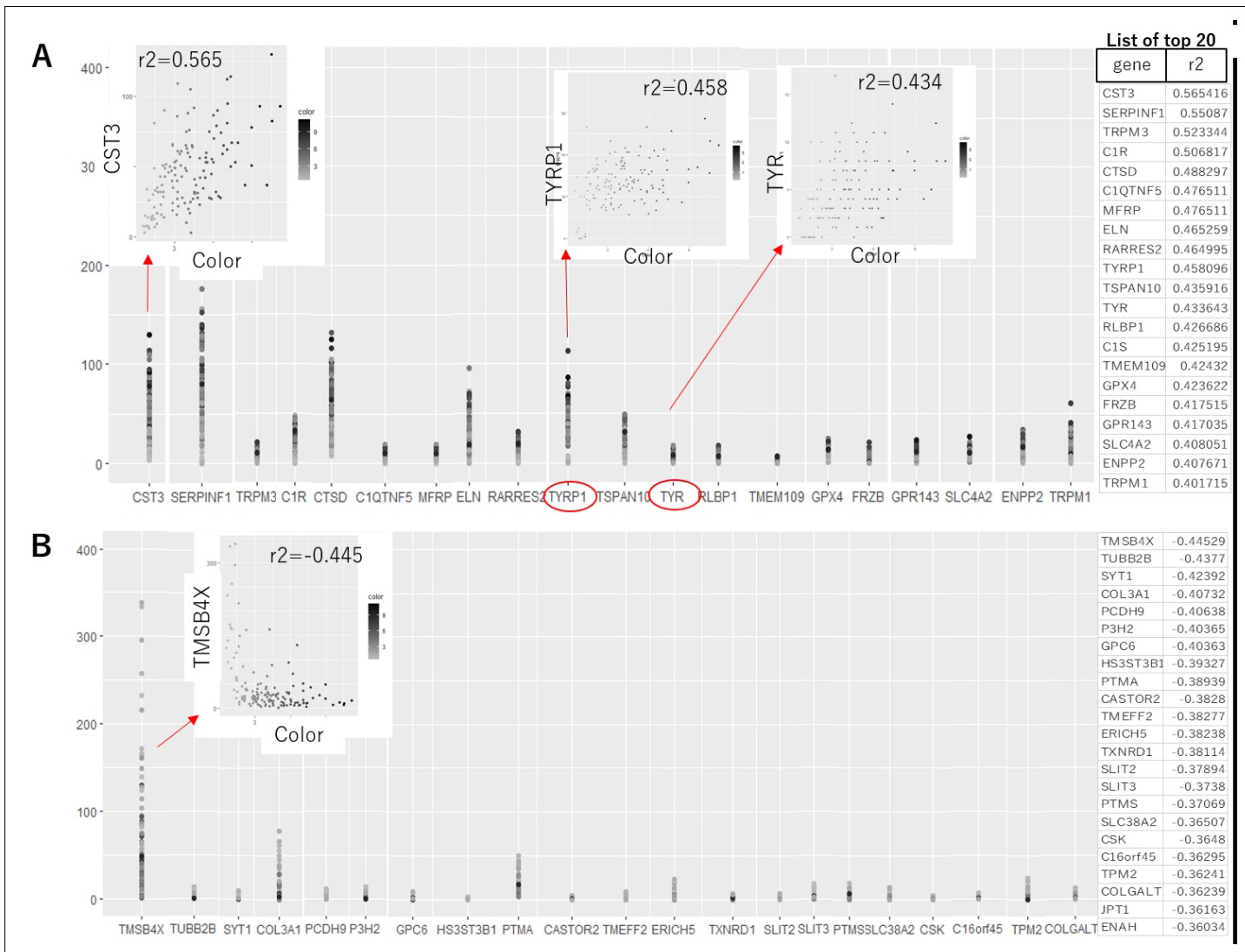

**Figure 3.** Correlation analysis between the expression level of each gene and the color of the human-induced pluripotent stem cell-derived RPE (iPSC-RPE). List of the genes with positive (**A**) and negative (**B**) correlations to color intensities of iPSC-RPE cells are shown. Even for the gene with the highest correlation to the color, the correlation value was 0.565. Notably, the melanin-synthesis enzymes *TYRP1* and tyrosinase (*TYR*) both showed only weak correlations (correlation value = 0.458 and 0.434, respectively). Program source code to create these plots is available in *Figure 3—source code 1*.

The online version of this article includes the following source code and figure supplement(s) for figure 3:

**Source code 1.** Program source code in R to create the plot shown in *Figure 3*.

**Figure supplement 1.** Hypothetical model for the interpretation of weak correlation between the expression of *TYR* and the degree of pigmentation in human-induced pluripotent stem cell-derived RPE cells (iPSC-RPE).

*and D*). These results indicated the degree of pigmentation of our iPSC-RPE did not associate with a specific gene expression profile defined by *t*-SNE.

## There was no strong correlation between the color of iPSC-RPE and the gene expression

As no correlation between the color and gene expression profile was revealed in our iPSC-RPE by clustering analysis, next we sought the correlation of individual genes with the color of the cells. When each of the 280,47 genes was analyzed for the correlation coefficient between its expression level and the color intensity in each cell, even for the gene with the highest positive- or negative-correlation with the color, the coefficient was 0.565 and –0.445, respectively (*Figure 3*). The gene with the highest correlation was *CST3* (correlation coefficient 0.565) that encodes cystatin C, a cysteine-proteinase inhibitor. As melanin, the material of the color, is cysteine-rich (D'Alba and *D'Alba and Shawkey, 2019*), it is reasonable to have increased level of melanin under upregulated *CST3* that inhibits cysteine-targeted degradation.

Intriguingly, the expressions of the genes directly related to the production of melanin were not the highest for the correlation with the color. For example, correlation coefficients between the expressions of the enzymes for melanin-synthesis, *TYR* and *TYRP1*, and color intensities were 0.434 and 0.458, respectively (*Figure 3A*). In other words, the darkest cells were not necessarily expressing the highest level of *TYR* or *TYRP1* mRNAs. This suggests the degree of pigmentation is dynamically regulated in each cell, and there is a time-lag between mRNA expression, production of the enzymes, and synthesis of melanin (descriptive image in *Figure 3—figure supplement 1*).

## Biological aspects that correlated with the pigmentation of iPSC-RPE

Having the results suggesting the color of iPSC-RPE may not be an indicator of the expression levels of RPE markers (*Figure 2—figure supplement 1*), next, we sought then which biological aspects underlie the degree of pigmentation of iPSC-RPE cells. For this purpose, we re-analyzed the transcriptome of our iPSC-RPE cells shown in *Figure 1A*. First, we calculated the weighted sum of the RGB channels, the so-called brightness, and used it as a proxy for the pigmentation level. Then, by gene set enrichment analysis (GSEA) (*Korotkevich et al., 2021*), we identified 15 biological pathways enriched in darker cells regardless of cell lineage (Lonza-RPE, iPS201-RPE, and iPS253-RPE), which included pathways involved in important functions the RPE, such as lysosome- and complement-related pathways (*Figure 4*). Among lysosome-related genes, *PASP* and *CTSD* correlated with the color intensity of the cells at correlation coefficients 0.34 and 0.31, respectively (*Figure 4—figure supplement 1*). Among complement-related genes, *C1R*, *C1S*, and *C3* correlated with the color intensity of the cells at correlation coefficients 0.31, 0.31, and 0.27, respectively (*Figure 4—figure supplement 2*). This correlation was partially consistent with the analysis of independent genes (*Figure 3*), which showed complement-related genes *C1R*, *C1QTNF5*, and *C1S*, and lysosome-related gene *CTSD* within the top 20 of the genes that had a positive correlation with color intensity (*Figure 3A* list on the right).

Gene set related to retinoid recycling, an important feature of the RPE to supply the photoreceptor with retinal, showed weak correlation with the color intensity of iPSC-RPE cells with correlation coefficients of genes such as *RDH11*, *BCO1*, *RDH10*, *DHRS3*, and *RDH5* at 0.25, 0.20, 0.16, 0.12, and 0.11, respectively (*Figure 4—figure supplement 3*). The gene set related to melanin synthesis, including the genes such as *DCT*, *TYRP1*, and *TYR*, showed less correlation with the color intensity of iPSC-RPE (*Figure 4—figure supplement 4*), which was consistent with the analysis of *TYRP1* and *TYR* shown in *Figure 3*.

## Discussion

In this study, we showed the degree of pigmentation of in vitro cultured iPSC-RPE cells did not project specific gene ontological clusters, although it correlated to some extent with the expressions of functional genes of the RPE, such as complement- and lysosome-related genes. Eye, as well as brain and testis, is an immune privilege organ, where inflammation by the immune system is minimized (*Zhou and Caspi, 2010*; *Chen et al., 2019*; *Mohan et al., 2022*). To protect the photoreceptors from pathogens in this immune suppressive environment, RPE has an immune cell-like aspect with the capability of complement activation (*Kunchithapautham et al., 2014*; *Chen et al., 2019*; *Jensen et al., 2020*;

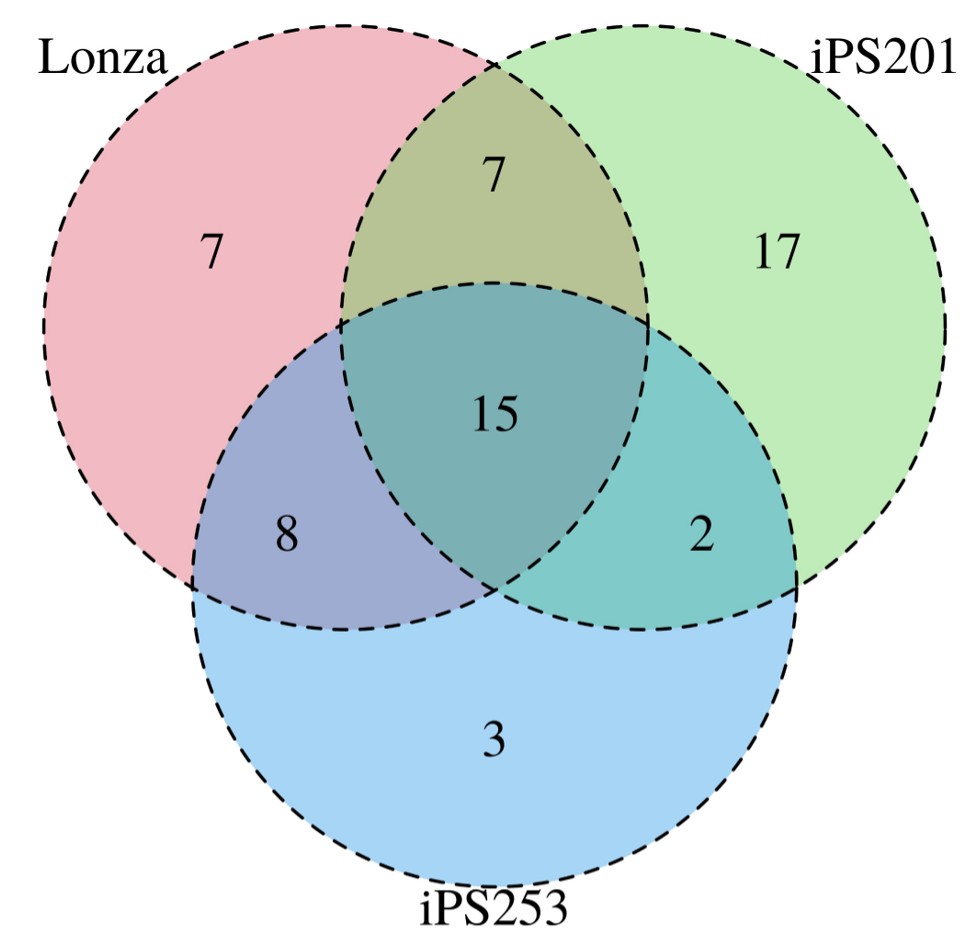

**KEGG_LYSOSOME**
KEGG_PROXIMAL_TUBULE_BICARBONATE_RECLAMATION
KEGG_PATHOGENIC_ESCHERICHIA_COLI_INFECTION
KEGG_SYSTEMIC_LUPUS_ERYTHEMATOSUS
KEGG_HYPERTROPHIC_CARDIOMYOPATHY_HCM
KEGG_ARRHYTHMOGENIC_RIGHT_VENTRICULAR_CARDIOMYOPATHY_ARVC
KEGG_DILATED_CARDIOMYOPATHY
KEGG_AXON_GUIDANCE
KEGG_FOCAL_ADHESION
KEGG_ECM_RECEPTOR_INTERACTION
**KEGG_COMPLEMENT_AND_COAGULATION_CASCADES**
KEGG_LEUKOCYTE_TRANSENDOTHELIAL_MIGRATION
KEGG_REGULATION_OF_ACTIN_CYTOSKELETON
KEGG_PATHWAYS_IN_CANCER
KEGG_SMALL_CELL_LUNG_CANCER

**Figure 4.** Gene set enrichment analysis (GSEA) of human-induced pluripotent stem cell-derived RPE (iPSC-RPE) cells elucidated gene sets correlated with color intensities. GSEA revealed 15 pathways that had a high correlation with the intensities of the color of iPSC-RPE. Among them, there were lysosome-related and complement-related pathways.

*Figure 4 continued on next page*

*Figure 4 continued*

The online version of this article includes the following figure supplement(s) for figure 4:

**Figure supplement 1.** Genes of lysosome-related pathway showed correlation with the color of iPSC-RPE cells.

**Figure supplement 2.** Genes of complement-related pathway showed moderate correlation with the color of human-induced pluripotent stem cell-derived RPE (iPSC-RPE) cells.

**Figure supplement 3.** Genes related to retinol metabolism had a weak correlation with the color of human-induced pluripotent stem cell-derived RPE (iPSC-RPE) cells.

**Figure supplement 4.** Genes related to melanogenesis had less correlation with the color of human-induced pluripotent stem cell-derived RPE (iPSC-RPE) cells.

*Tan et al., 2020*; *Schäfer et al., 2020*) and phagocytosis (*Boulton, 2014*; *Lehmann et al., 2014*). Although it was weak, the correlation of complement- and lysosome-related genes to the color shown in the present study, suggests the darker RPE cells may be prone to facilitate these functions.

Besides these immunogenic aspects, pigmentation also confers a protective function on RPE from different biological aspects. The material of the dark pigments, melanin, is conserved from bacteria to mammals, with diverse protective functions against damaging ultraviolet rays, free radicals, or toxins (*D'Alba and Shawkey, 2019*). In most species, melanin pigments are confined in an intracellular organelle called melanosome. In the eye, melanosomes are located in RPE cells and choroidal melanocytes, where they shield the photoreceptor to reduce backscattered light and remove free radicals (*Boulton, 2014*; *Lehmann et al., 2014*; *D'Alba and Shawkey, 2019*). Unlike skin melanin that is constantly synthesized in epidermal melanocytes, melanin in RPE cells decreases with age, making the retina vulnerable with less protection by the RPE. Several attempts were made to re-pigment RPE cells by supplying them with melanosomes isolated from ex vivo RPE cells (*Boulton, 2014*) or more recently with artificial melanin-like nanoparticles (*Kwon et al., 2022*). In accordance with decreased amount of melanin in adult RPE, primary cultured adult RPE cells are less pigmented (*Boulton, 2014*). On the other hand, primary culture of fetal-derived RPE cells, although less pigmented initially, becomes heavily pigmented when confluent monolayers are formed (*Maminishkis et al., 2006*), reflecting the nature of melanosomes produced during a limited time window of embryogenesis (*Boulton, 2014*). More importantly, human embryonic stem cell (ESC)- or iPSC-derived RPE cells exhibit pigmentation when they become confluent (*Boulton, 2014*; *Kamao et al., 2014*) suggesting these cells may retain or have been reprogrammed to gain the characteristics of fetal RPE cells. The Lonza-RPE cells (line #476621) used in this study were fetal-derived. Our iPSC-RPE formed different clusters from Lonza-RPE by single-cell transcriptome analysis, which was consistent with our previous study showing the gene expression pattern of our iPSC-RPE was slightly different from Lonza-RPE, although it was closer than human RPE cell-line ARPE19, (*Kamao et al., 2014*). Interestingly, the pigmentation level of Lonza-RPE cells correlated with their gene expression profile (*Figure 2D*), which may reflect the developmental process of RPE that gains protective function, including melanogenesis, during embryogenesis. From this aspect, ESC- or iPSC-derived RPE cells may be more plastic, retaining immature profiles although apparently being pigmented spontaneously. In fact, there are several lines of evidence implying in vitro pigmentation of stem cell-derived RPE cells may not necessarily reflect their levels of functional maturation. For example, bone marrow-derived RPE cells that are poorly pigmented in vitro become highly pigmented when transplanted (*Sengupta et al., 2009*), as well as our iPSC-RPE cells that are not pigmented when prepared but actually become heavily pigmented after engrafted (*Mandai et al., 2017*; *Sugita et al., 2020*), suggest the significance of environmental niche for melanogenesis. When stem cell-derived RPE cells are cultured in vitro, they are apparently not pigmented at sparse density, probably due to dilution amongst daughter cells (*Boulton, 2014*) or secretion of melanin by the fusion of melanosome membrane to the plasma membrane as proposed in melanocytes (*Moreiras et al., 2021*), but they become pigmented when they form a confluent monolayer, which is a reversible effect as the dynamics of pigmentation repeats when they are re-plated at a sparse density and become confluent again (*Figure 3—figure supplement 1*). This again suggests the involvement of extracellular cues for pigmentation, besides the intrinsic characteristics of each RPE cell. Indeed, it has been shown that in vitro pigmentations of RPE cells are enhanced by extracellular matrix (*Boulton, 2014*) or even more intriguingly by phagocytosis of rod outer segments (*Schraermeyer et al., 2006*).

Without the exposure to pathogens or photoreceptor outer segments, in vitro pigmentation of iPSC-RPE cells could be somewhat a spontaneous but not a necessary sign to show they are prone to execute their protective function.

## Materials and methods

**Key resources table**

| Reagent type (species) or resource | Designation | Source or reference | Identifiers | Additional information |
|---|---|---|---|---|
| Cell line (human) | Human iPSC 201B7 (HPS4290) | RIKEN BRC / National BioResource Project of the MEXT/AMED, Japan | | doi:10.1038/nbt1374 |
| Cell line (Homo sapiens) | Human iPSC 253G1 (HPS0002) | RIKEN BRC / National BioResource Project of the MEXT/AMED, Japan | | doi:10.1038/nbt1374 |
| Cell line (Homo sapiens) | Human embryonic RPE (line #476621) | Lonza | Lonza: #476621 | |
| Cell line (Homo sapiens) | Human dermal fibroblasts | doi: 10.1167/iovs.14-15619 | TLHD1 | obtained from a healthy male donor |
| Commercial assay or kit | Bravo NGS workstation | Agilent Technologies | | |
| Software, algorithm | Seurat v4 | doi:10.1038/nbt.3192; doi:10.1038/nbt.4096; doi:10.1016/j.cell.2019.05.031; doi:10.1016/j.cell.2021.04.048 | | |
| Software, algorithm | R v4.2.1 | R Development Core Team, 2022 | | |
| Software, algorithm | weighted sum of the background-corrected intensities | International Telecommunication Union, 2022 | | |
| Software, algorithm | Gene Set Enrichment Analysis; GSEA-preranked | GenePattern.org | | |

### Cell culture

The study was approved by the ethical committees of the Institute of Biomedical Research and Innovation Hospital and the RIKEN Center for Developmental Biology, Japan. Human iPSC-lines 201B7 (HPS4290) and 253G1 (HPS0002) (*Nakagawa et al., 2008*) were provided by the RIKEN BRC through the National BioResource Project of the MEXT/AMED, Japan. iPSCs were cultured and differentiated into RPE cells as described previously (*Kamao et al., 2014*). Briefly, to differentiate human iPSCs into RPE cells, human iPSCs were cultured on gelatin-coated dishes in differentiation medium (GMEM (Sigma-Aldrich, St. Louis, MO) supplemented with 1 mM sodium pyruvate, 0.1 mM non-essential amino acids (Sigma-Aldrich), and 0.1 mM 2-mercaptoethanol (Sigma-Aldrich)) with 20% KnockOut Serum Replacement (KSR; Invitrogen, Waltham, MA) for four days, 15% KSR for six days, and 10% KSR for 20 days. Y-27632 (10 μM; FUJIFILM Wako, Osaka, Japan), SB431542 (5 μM; Sigma-Aldrich), and CKI7 (3 μM; Sigma-Aldrich) were added for the initial 18 days. After the emergence of pigmented cells, the medium was switched to SFRM (DMEM/F12 [7:3] supplemented with B27 (Invitrogen), 2 mM L-glutamine).

Lonza-RPE cells (line #476621; Lonza, Basel, Switzerland) were maintained in SFRM as well.

Human dermal fibroblasts were obtained from a healthy donor and cultured as described previously (*Sugita et al., 2015*).

### Cell preparation for ALPS

Cell concentrations was measured using Hemocytometer Standard Specification (Improved Neubauer) (HIRSCHMANN LAB, Baden-Württemberg, Germany). 3 ml of PBS (D-PBS(-) without Ca and Mg, #14249–95, Nacalai Tesque, Kyoto, Japan) containing $0.25–1.0×10^4$ cells were added onto a dish (PrimeSurface Dish 35, #MS-9035X, Sumitomo Bakelite, Tokyo, Japan).

### Imaging, isolation, and RNA-seq for single-cells

Live cell imaging, cell picking, and single-cell digital RNA-seq (*Shiroguchi et al., 2012*; *Ogawa et al., 2017*) were performed as described previously (*Jin et al., 2023*) except for the following: In total, 4032 cells (*Figure 1—source data 1*) were measured and analyzed. Cell images (400 pixels ×330

pixels; 0.36 µm/pixel) were captured by both bright field and fluorescent channels (filter unit: mCherry C-FLL-C, Nikon Co., Tokyo, Japan) with 20X objective (N.A. 0.7) and a color camera (Color Camera Nikon DS-Ri2, Nikon Co.). Exposure times for bright field and fluorescence were 10 ms and 100 ms, respectively. For both channels, Z-stacks (21 planes) were recorded at 1 µm interval. Randomly selected cells were picked ('random selection' of ALPS *Jin et al., 2023*). For the picked single cells, library preparation including cell lysis, RNA fragmentation, cDNA generation with molecular barcode attachment, amplification, and purification was performed using the Bravo NGS workstation (Agilent Technologies, Santa Clara, CA) and thermal cyclers (Mastercycler X50s; Eppendorf, Hamburg, Germany). The libraries were mixed using different indexes, and sequenced on HiSeq (150 cycle; Illumina, San Diego, CA) using custom primers. Detected number of RNA molecules for each gene in each cell was counted based on molecular barcodes.

To estimate the RGB values of the cells within the cell images, first, the average intensities (0–255) per pixel for R, G, and B channels, respectively, of a circular area with a radius of 50 pixels at the position (195, 169 (coordinates in pixel)) of each cell image that covered the cell, and the area other than the circular area (background) were calculated. Then, for each image and each channel, the average intensity of the background was subtracted from the average intensity of the circular area. Finally, for each cell and each channel, the mean and standard deviation of the subtracted average circular area intensities of all 21 Z-stack images were obtained. The circular area images of iPSC-RPE cells are illustrated in *Figure 1B*.

## Single-cell transcriptome analysis

Single-cell RNA-seq data were analyzed using Seurat v4 (*Satija et al., 2015*; *Butler et al., 2018*; *Stuart et al., 2019*; *Hao et al., 2021*) with R v4.2.1 (*R Development Core Team, 2022*). For cluster analyses in *Figure 2*, the following parameters were used:

1. FindNeighbors(dims = 1:6)
2. FindClusters(resolution = 0.4)
3. RunTSNE(all, dims = 1:6).

## Brightness calculation

The brightness value $Y$ for each cell was calculated by taking a weighted sum of the background-corrected intensities of the RGB channels (*International Telecommunication Union, 2022*):

$$Y = (0.299, \, 0.587, \, 0.114) \cdot (R_{\text{corr}}, \, G_{\text{corr}}, \, B_{\text{corr}}),$$

where the background-corrected intensities of the RGB channels were obtained by adding a virtual white background to delta intensities of RGB channels of each cell:

$$(R_{\text{corr}}, G_{\text{corr}}, B_{\text{corr}}) = (255, \, 255, \, 255) + (\Delta R, \Delta G, \Delta B).$$

If the corrected intensity exceeded 255, it was replaced by the maximum value below 255 (there was only one such case). The resultant cellular brightness $Y \in [0, 255]$ had a skewed distribution around the background brightness value, 255. Then, normalized brightness $Y_{\text{norm}}$ was obtained by logit transformation, so that its range takes the entire real number.

$$Y_{\text{norm}} = \log \left( \frac{Y}{255 - Y} \right)$$

The normalized brightness $Y_{\text{norm}} \in [-\infty, \infty]$ was approximately normally distributed.

## Pathway enrichment analysis

For each of the three RPE cell lineage (Lonza-RPE, iPS201-RPE, and iPS253-RPE), we explored biological pathways correlated with pigmentation levels according to the following steps:

1. Calculate Pearson's correlation coefficient between scaled gene expression levels ('scale.data' slot of a Seurat object) and the normalized brightness $Y_{\text{norm}}$.

2. Perform GSEA-preranked by the fgsea library in R using the gene list ranked by absolute value of Pearson's correlation coefficient as input and the KEGG pathway of MSigDB (c2.cp.kegg. v2023.1.Hs.symbols.gmt) as a reference (*Liberzon et al., 2011*; *Korotkevich et al., 2021*).
3. Define pathways with FDR <0.05 as significant (conventional criteria for GSEA).
4. Divide significant pathways into brightness-correlated pathways (NES >0) and -uncorrelated pathways (NES <0).

Finally, for each of the brightness-correlated and -uncorrelated significant pathways, we extracted the intersection, i.e., pathways commonly enriched in all three cell lineages. Note that although we found 15 common brightness-correlated pathways among three cell lines, no brightness-uncorrelated pathways were detected in any of the three cell lines.

## Additional information

### Competing interests
Yoko Nakai-Futatsugi, Noriko Sakai, Seiji Hori: Employee of VC Cell Therapy Inc. Ken-ichi Hironaka, Masakazu Fukuda, Hiroki Danno: Employee of Knowledge Palette, Inc. Masayo Takahashi: Employee of Vision Care, Inc. The other authors declare that no competing interests exist.

### Funding

| Funder | Grant reference number | Author |
|---|---|---|
| Japan Society for the Promotion of Science | 18H05411 | Katsuyuki Shiroguchi |
| RIKEN | Special Postdoctoral Researchers Program of RIKEN | Taisaku Ogawa |

The funders had no role in study design, data collection and interpretation, or the decision to submit the work for publication.

### Author contributions
Yoko Nakai-Futatsugi, Conceptualization, Data curation, Software, Formal analysis, Investigation, Visualization, Methodology, Writing – original draft; Jianshi Jin, Resources, Data curation, Software, Formal analysis, Validation, Investigation, Visualization, Methodology, Writing – review and editing; Taisaku Ogawa, Conceptualization, Resources, Data curation, Software, Formal analysis, Funding acquisition, Investigation, Visualization, Methodology; Noriko Sakai, Resources, Data curation, Investigation; Akiko Maeda, Conceptualization, Resources, Formal analysis, Writing – review and editing; Ken-ichi Hironaka, Conceptualization, Data curation, Software, Formal analysis, Validation, Investigation, Visualization, Methodology, Writing – review and editing; Masakazu Fukuda, Conceptualization, Software, Formal analysis, Validation, Visualization, Methodology, Writing – review and editing; Hiroki Danno, Conceptualization, Software, Formal analysis, Supervision, Methodology, Project administration, Writing – review and editing; Yuji Tanaka, Conceptualization, Data curation, Investigation, Methodology; Seiji Hori, Masayo Takahashi, Conceptualization, Supervision, Project administration; Katsuyuki Shiroguchi, Conceptualization, Resources, Data curation, Software, Formal analysis, Supervision, Funding acquisition, Investigation, Methodology, Project administration, Writing – review and editing

### Author ORCIDs
Yoko Nakai-Futatsugi ⓘ https://orcid.org/0000-0003-4187-6060
Seiji Hori ⓘ https://orcid.org/0009-0009-7282-4838

Reviewer #2 (Public review): https://doi.org/10.7554/eLife.92510.3.sa1
Reviewer #3 (Public review): https://doi.org/10.7554/eLife.92510.3.sa2
Author response https://doi.org/10.7554/eLife.92510.3.sa3

## Additional files

### Supplementary files
• MDAR checklist

### Data availability

All RNA-sequencing datasets created in this study have been deposited in the Gene Expression Omnibus database under accession number GSE242184.

The following dataset was generated:

| Author(s) | Year | Dataset title | Dataset URL | Database and Identifier |
|---|---|---|---|---|
| Nakai-Futatsugi Y, Jin J, Ogawa T, Sakai N, Maeda A, Hironaka K, Fukuda M, Danno H, Tanaka Y, Hori S, Shiroguchi K, Takahashi M | 2023 | Pigmentation level of human iPSC-derived RPE does not indicate a specific gene expression profile | http://www.ncbi.nlm.nih.gov/geo/query/acc.cgi?acc=GSE242184 | NCBI Gene Expression Omnibus, GSE242184 |

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
