## [Editor Report · eLife assessment]

This **useful** work describes a novel microscopy-based method to correlate the degree of pigmentation with the gene expression profile of human-induced pluripotent stem cell-derived Retinal Pigmented Epithelial (iPSC-RPE) cells at the single cell level. The presented evidence is **solid** in showing that there is heterogeneous gene expression in iPSC-derived RPE cells, and there is no significant correlation with the pigmentation. By analyzing the expression of some genes related to function, lysosomal- and complement-related pathways were partially enriched in darker cells. This methodology can be used by other researchers interested in analyzing gene expression related to microscopic images.

---

## [Referee Report · Reviewer #2 (Public review)]

Summary:

in this paper authors show that the degree of pigmentation for RPE cells is not correlated with a level of maturation and function. They suggest that this status could be different in vitro than in vivo but do not provide proper experiments to validate this hypothesis. However, it is the first time that the absence of correlation between pigmentation and function is studied.

Strengths:

The methods are good and experiments very rigorous

Comments on current version:

The authors have modified their title and focus on QC for in vitro process

---

## [Referee Report · Reviewer #3 (Public review)]

Summary:

Nakai-Futatsugi et al. present a novel method to analyze the correlation between the degree of pigmentation and the gene expression profile of human-induced pluripotent stem cell-derived RPE (iPSC-RPE) cells at the single cell level. This was achieved with the use of ALPS (Automated Live imaging and cell Picking system), an invention developed by the same authors. Briefly, it allows one to choose and photograph a specific cell from a culture dish and proceed to single cell digital RNA-seq. The authors identify clusters of cells that present differential gene expression, but this showed no association with the degree of pigmentation of the cells. Further data analysis allowed the authors to correlate the degree of pigmentation to some degree with the expression of complement and lysosome-related genes.

Strengths:

An important amount of data related to gene expression and heterogeneity of the iPSC-RPE population has been generated in this work.

Weaknesses:

However, the justification of the analysis, and the physiological relevance of the hypothesis and the findings could be strengthened.

Importantly, I fail to grasp from the introduction what is the previous evidence that leads to the hypothesis. Why would color intensity be related to the quality of cell transplantation? In fact, cell transplantation is not evaluated at all in this work. The authors mention "quality metrics for clinical use", but this concept is not further explained. Neither is the concept of "sufficient degree of pigmentation" explained.

On the other hand, the positive correlation of cluster formation with complement and lysosome-related genes is not discussed.

As a consequence it is very difficult to evaluate the impact of these findings on the field.

---

## [Author Response]

The following is the authors’ response to the original reviews.

We thank the reviewers for their careful comments. We sincerely agree with the comments from both reviewers, and noticed the word “cell transplantation”, throughout the manuscript including the title, was confusing. We revised the manuscript to clarify the aim of the study, and to express the conclusion more straightforwardly.

Response to the reviewers:

We interpret the data of the present study as the color of each RPE cell is a temporal condition which does not necessarily represent the quality (e.g. for cell transplantation) of the cells. We consider this may be applicable not only in vitro but also in vivo, although we do not know whether RPE shows heterogeneous level of pigmentation in vivo.

As our concern for iPSC-RPE is always about their quality for cell transplantation, maybe we haven’t fairly evaluated the scientific significance obtained from the present study.

Another thing we noticed was, although we used the term “cell transplantation” to explain what we meant by “quality” of the cells, we agree this was confusing. The aim of the study was not to show how the pigmentation level of transplant-RPE affects the result of cell transplantation, but to show the heterogeneous gene expression of iPSC-derived RPE cells, and the less correlation of the heterogeneity with pigmentation level. We went through the manuscript, including the title, to more straightforwardly lead this conclusion: the degree of pigmentation had some but weak correlation with the expression levels of functional genes, and the reason for the weakness of the correlation may be because the color is a temporal condition (as we interpreted from the data) that is different from more stable characteristics of the cells.

We agree that “cell transplantation” in the title (and other parts) was misleading. So, we changed the title, and removed the phrase that led as if the aim of the study was to show something about cell transplantation or in vivo results.

Also, to face scientifically significant results obtained from the present study appropriately, we discussed more about the correlation of the pigmentation level with some functional genes, and brought this as one of the conclusions of the manuscript.